# The Effect of Different Types of Fertilizers on the Growth of Cassava and the Fungal Community in Rhizosphere Soil

**DOI:** 10.3390/jof11030235

**Published:** 2025-03-19

**Authors:** Qinyun Liu, Xiaojing Lu, Chunyu Xiang, Shan Yu, Jie Zhang, Kaimian Li, Wenjun Ou, Songbi Chen, Jie Cai

**Affiliations:** 1Tropical Crops Genetic Resources Institute, Chinese Academy of Tropical Agricultural Sciences/Key Laboratory of Ministry of Agriculture for Germplasm Resources Conservation and Utilization of Cassava, Haikou 571101, China; liuqinyun1994@163.com (Q.L.); lu104@163.com (X.L.); 13698979213@163.com (S.Y.); zj772277@163.com (J.Z.); likaimian@sohu.com (K.L.); cassava6973@163.com (W.O.); 2School of Life and Health Sciences, Hainan University, Haikou 570228, China; 18896171787@163.com; 3Sanya Research Institute, Chinese Academy of Tropical Agricultural Sciences, Sanya 572024, China; 4Key Laboratory of Tropical Crops Germplasm Resources Genetic Improvement and Innovation of Hainan Province, Haikou 571101, China

**Keywords:** cassava, different fertilizers applied, soil properties and enzyme activities, fungal structure communities, cassava growth

## Abstract

With the growing importance of cassava worldwide, developing efficient and eco-friendly fertilizer strategies is crucial for sustainable cassava production. Diverse fertilizer treatments can significantly influence soil properties and plant growth. In this study, we investigated the effects of three fertilizer treatments—organic fertilizer (OF), chemical fertilizer combined with organic fertilizer (CFOF), and reduced chemical fertilizer combined with organic fertilizer (RFOF)—on the fungal community structure, chemical properties (SOM, AP, AN, and AK), and enzyme activities (NP, SC, CAT, and UE) in cassava rhizosphere. Our results demonstrated that these fertilizer treatments significantly enhanced cassava growth and yield compared to the control (CK) without fertilization. Soil chemical properties (SOM, AN, AP, and AK) and enzyme activities (NP, SC, CAT, and UE) were notably improved following fertilization. High-throughput sequencing revealed the significant alterations in the relative abundance of specific fungal taxa. Environmental parameters, particularly UE, SC, CAT, and AP, showed strong correlations with fungal community structure. These findings highlight the critical role of combined organic and chemical fertilizers in promoting cassava productivity and soil health. Understanding these interactions provides a foundation for optimizing fertilization practices to enhance crop yields and support sustainable agriculture.

## 1. Introduction

Crop production depends on healthy soil to maintain crop growth and yield. Healthy soil provides essential nutrients, water, air, and temperature while also supporting root growth and development. The term ‘soil health’ is used to describe the capacity of soil to function within ecosystem boundaries in a manner that sustains crop and animal productivity, maintains or enhances environmental sustainability, and improves global human health [1]. The key indicators of soil health are the biological, physical, and chemical properties of the soil, which collectively influence soil functions. Traditionally, chemical properties, such as pH, organic matter, and nutrient content, have been used as soil health indicators [2]. Soil parameters that provide information on the biomass, enzyme activity, and diversity of soil microorganisms have been employed as bio-indicators of soil health [3,4,5,6]. The rhizosphere microbiome, another important indicator of soil health, also recognized as the second genome of plants, plays an important role in agricultural ecosystems [7,8]. There is substantial evidence to suggest that these microorganisms affect the health, growth, and environmental adaptability of their host plants [9]. Recent studies have demonstrated that different fertilizations have impact on the structure of the fungal community [10,11,12].

Due to the increasing demand for food, animal feed, and biofuels, global crop production is currently being expanded [13]. Currently, 47.8 million km^2^ of land are used for agriculture, which corresponds to around 50% of the world’s habitable land [14]. Fertilizers are still required because nutrients may become insufficient in soil. However, regarding the efficiency of these fertilizers, particularly in the case of nitrogen (N), only 7–58% of applied nitrogen is typically absorbed by plants across diverse tropical agroecosystems [15]. Meanwhile, the rest of the fertilizer runs off into water bodies, resulting in a range of adverse environmental consequences, including water contamination, soil acidification, and increased emission of greenhouse gases [16,17]. The impact of fertilizers on cropping systems has predominantly been investigated in relation to crop yield and their environmental outcomes [18,19,20,21]. In contrast to synthetic chemical fertilizers, organic fertilizers are released at a slower rate, which is eco-friendly. Moreover, they can also be a possible source of soil microbes [22]. One such example is Lignite Bioorganic Fertilizer (LBF), which demonstrates significant advantages in improving soil quality, reducing greenhouse gas emissions, enhancing carbon sequestration, promoting nitrogen cycling, increasing crop yields, and offering substantial economic and environmental benefits [23]. These multifaceted benefits make LBF a promising solution for sustainable agricultural practices and climate change mitigation. However, the slow release of organic fertilizers also presents a challenge, making them insufficient to meet the nutritional needs of certain crops at a specific point in time [24]. Considering the above, reduced chemical fertilizer application and the integration of organic fertilizers represent promising strategies to mitigate environmental risks while maintaining or even improving crop yields [24,25,26,27,28].

Cassava (*Manihot esculenta* Crantz) is a vital staple crop for millions of people, particularly in tropical and subtropical regions [29]. The species demonstrates a robust capacity to adapt to diverse environmental conditions, particularly in regions where soil is infertile, or rainfall is unpredictable [30]. In the cultivation of cassava, mineral fertilizers are routinely applied to ensure better growth and boost root productivity. The positive effects of different fertilizers on cassava growth, quality, and yield have been the subject of several studies [31,32]. The effects of integrating organic (chicken manure) and inorganic (NPK) fertilizers on cassava growth, yield, and nutrient use efficiency in two agroecological zones of Zambia were assessed, with findings showing that combined applications significantly enhance productivity and agronomic efficiency compared to sole NPK fertilizer use [33]. However, the underlying molecular mechanisms remain unknown. A limited number of studies have investigated the abundance and diversity of soil microorganisms in cassava rhizomes following the application of fertilizers [11,32]. Nevertheless, there is currently no evidence in the literature concerning the impact of combining different fertilizers with organic fertilizer application on the structure of fungal communities in cassava rhizosphere soils. With the growing importance of cassava worldwide, there is increasing demand for a more effective and ecofriendly cultivation method.

The objective of this study was to investigate the impact of different fertilizers, especially a reduced chemical fertilizer application in conjunction with organic fertilizer, on soil chemical properties, soil fungal communities, plant growth parameters, and yield in cassava rhizosphere. By identifying environmentally sustainable fertilization strategies tailored to cassava production systems, we aim to contribute to the advancement of resilient and ecologically sound agricultural practices in the context of mounting environmental challenges.

## 2. Material and Methods

### 2.1. Description of Study Site and Materials

The experiment site was located at the National Cassava Germplasm Repository (NCGR) at Danzhou, Hainan, Province, China, in 2022-2023 (19°30′ N, 109°30′ E). The cassava cultivar South China 12 (SC12) was provided by NCGR for cultivation in this study. The chemical fertilizers used in this study were urea (N > 46%), calcium superphosphate (P_2_O_5_ > 16%), and potassium sulfate (K_2_O > 52%). The organic fertilizer (N+P_2_O_5_+K_2_O ≥ 5%, organic matter ≥ 80%, total humic acid ≥ 30 g/kg, humus ≥ 300 g/kg), derived from lignite fermentation, was provided by Yuan Tai Feng (Baotou) Biotechnology Co., Ltd., Baotou, China.

### 2.2. Experimental Design and Sample Collection

In this study, four different treatments were set (Table 1). The CK involved no application of any fertilizers and served as the control group. The CFOF involved the application of conventional chemical fertilizers and organic fertilizer, specifically, N 90 kg ha^−1^, P_2_O_5_ 90 kg ha^−1^, K_2_O 90 kg ha^−1^, and organic fertilizer 1500 kg ha^−1^. The RFOF involved the application of reduced chemical fertilizer (reducing nitrogen by 25% and phosphorus by 50% compared to conventional fertilization) and organic fertilizer combination, specifically, N 67.5 kg ha^−1^, P_2_O_5_ 45 kg ha^−1^, K_2_O 90 kg ha^−1^, and organic fertilizer 1500 kg ha^−1^. The OF involved the application of 1500 kg ha^−1^ organic fertilizer only.

Each experimental plot had an area of 36 m^2^, with three replications, totaling 12 experimental plots. Each plot was planted with 6 rows, with 6 plants per row, totaling 36 plants per plot, with a spacing of 1.0 m × 1.0 m between plants. The distance between plots was 50 cm, and the distance between replications was 100 cm. Fertilizers were applied once at a depth of 15–30 cm from the base of the cassava stem two months after planting. Based on the plant density (10,000 plants per hectare) and the application rates, the exact amount of each fertilizer applied per plant was calculated. For the OF, each plant received 4.17 g of organic fertilizer without any chemical fertilizer. For the CFOF, each plant received 9 g urea, 9 g calcium superphosphate, and 9 g potassium sulfate, along with 4.17 g of organic fertilizer. For the RFOF, each plant received 6.75 g urea, 4.5 g calcium superphosphate, and 9 g potassium sulfate, along with 4.17 g of organic fertilizer. SC12 was planted in May 2022 and harvested in March 2023.

Cassava rhizosphere soil samples were collected in March 2023, following the methods described by Cai et al. [11]. All tools, collection bags, or other items used during sampling were sterilized. Ten sampling points were established, and samples were collected in an S-shaped pattern. Sterile brushes were used to brush the soil adhering to the surface of cassava roots onto newspapers. All samples were then thoroughly mixed after passing through a 1 mm soil sieve to remove debris, constituting the rhizosphere soil sample for each experimental plot. After labeling the collected cassava soil samples, one portion of collected sample was used for soil chemical property and enzyme activity detection. Another portion was placed in dry ice for subsequent high-throughput sequencing analysis.

### 2.3. Soil Chemical Property Analysis

The chemical properties of the soil were analyzed using the methods described by Lu et al. [34]. Specifically, the soil organic matter (SOM) content was quantified through titration following oxidation with K_2_Cr_2_O_7_. The alkaline dissolved nitrogen (AN) concentration was determined through the diffusion method. The available phosphorus (AP) was quantified through the hydrochloric acid–ammonium fluoride method. The available potassium (AK) concentration was determined through ammonium acetate extraction. (NP, SC, CAT, and UE).

### 2.4. Soil Enzyme Activity Detection

The determination of soil enzyme activities was performed using the colorimetric methods proposed by Gao et al. [35]. Urease (UE) activity was measured using the indigo carmine colorimetric method. Briefly, soil samples were incubated with urea, and the hydrolysis of urea to ammonia was detected by the formation of indigo carmine dye in an alkaline solution. The absorbance was measured at 710 nm using a spectrophotometer. Sucrase (SC) activity was determined using the 3,5-dinitrosalicylic acid (DNS) colorimetric method. Soil samples were incubated with sucrose, and the reducing sugars produced were reacted with DNS reagent. The absorbance was measured at 540 nm. Neutral phosphatase (NP) content was assessed with the *p*-nitrophenyl phosphate (*p*-NPP) colorimetric method. Soil samples were incubated with *p*-NPP, and the release of *p*-nitrophenol was measured at 410 nm. Catalase (CAT) content was evaluated using a spectrophotometric method based on the decomposition of hydrogen peroxide. The rate of oxygen release was monitored by measuring the absorbance at 240 nm.

### 2.5. Parameters of Cassava Growth and Yield

Stem girth, plant height, number of tuberous roots, and yield were registered and analyzed as described by Cai et al. [11]. Plant height was measured from the base of the cassava stem to the tip of the tallest leaf using a graduated measuring tape before harvest. Stem girth was measured at a height of 30 cm from the base of the cassava stem using a flexible measuring tape. The number of tuberous roots per plant was determined by manually counting the number of harvestable tubers after uprooting each plant. After harvesting, all tubers from each plot were weighed using a calibrated electronic scale.

### 2.6. Soil DNA Extraction, Amplicon Generation, and Sequencing

The MOBIO Power Soil^®^ DNA Isolation Kit (MOBIO Laboratories, Carlsbad, CA, USA) efficiently extracts soil DNA by leveraging the selective binding properties of silica membranes. The concentration and purity were measured using the NanoDrop One (Thermo Fisher Scientific, Waltham, MA, USA). For the PCR reactions system: 25 μL 2 × Premix Taq (Takara Biotechnology, Dalian Co., Ltd., Dalian, China), 1 μL of each primer (ITS3′-F 5′-TCCTCCGCTTATTGATATGC-3′, ITS4′-R 5′-TCCTCCGCTTATTGATATGC-3′; 10 μM), and 3 μL DNA (20 ng/μL) template in a volume of 50 μL were amplified by thermocycling: 5 min at 94 °C for initialization; 30 cycles of 30 s denaturation at 94 °C, 30 s annealing at 52 °C, and 30 s extension at 72 °C. This was followed by 10 min final elongation at 72 °C. The PCR instrument was BioRad S1000 (Bio-Rad Laboratory, Hercules, CA, USA). The length and concentration of the PCR product were detected by 1% agarose gel electrophoresis. Sequencing libraries were generated using NEBNext^®^ Ultra™ II DNA Library Prep Kit for Illumina^®^ (New England Biolabs, Ipswich, MA, USA) following the manufacturer’s recommendations, and index codes were added. The library quality was assessed on the Qubit@ 2.0 Fluorometer (Thermo Fisher Scientific, Waltham, MA, USA). At last, the library was sequenced on an Illumina Nova6000 platform and 250 bp paired-end reads were generated (Guangdong Magigene Biotechnology Co., Ltd., Guangzhou, China).

### 2.7. Sequencing Data Analysis

High-throughput sequencing of the ITS region was performed by Guangdong Magigene Genomics Co., Ltd., Guangzhou, China to profile fungal communities. Raw sequencing data underwent a rigorous quality control process to remove low-quality reads, adapter sequences, and chimera sequences, resulting in high-quality clean reads suitable for downstream analysis.

All bioinformatics analyses were conducted using a combination of USEARCH (version V10) and R software (version 4.0.3), leveraging the Magigene Cloud platform (http://cloud.magigene.com/) for streamlined data processing and analysis. Operational taxonomic unit (OUT) clustering was conducted using USEARCH with a similarity threshold of 97%, and alpha diversity indices (including Shannon, Simpson, Observed-species, Chao1, ACE, and Good-coverage) were calculated using the usearch_alpha_div function. Beta diversity analysis was performed using the vegan package (version 2.5-3) in R to assess community structure differences among samples. Principal coordinate analysis (PCoA) based on Bray–Curtis dissimilarity was used to visualize sample clustering, and a PERMANOVA was conducted using the adonis function to evaluate the significance of community differences across treatments. To explore the relationships between microbial community structure and environmental factors, canonical correspondence analysis (CCA) was performed using the cca function in the vegan package, with environmental vectors added using the envfit function to illustrate significant correlations (r^2^ and *p*-values). Hierarchical clustering of sample distances was visualized using heatmaps generated by the pheatmap package (version 1.0.12), and Venn diagrams were constructed using the VennDiagram package (version 1.7.3) to analyze the overlap and uniqueness of OTUs among different treatments.

### 2.8. Statistical Analysis

Data processing and table generation were performed using Microsoft Excel 2003. Subsequent statistical analyses were carried out using GraphPad Prism 9 to evaluate significant differences among treatments and to investigate relationships between variables.

To compare multiple groups and identify significant differences, the Kruskal–Wallis H-test was utilized. This non-parametric test is appropriate for datasets that do not meet the assumptions of normality or homogeneity of variances. The Kruskal–Wallis test was performed with a significance threshold of *p* < 0.05. When significant differences were detected, pairwise comparisons were conducted using Dunn’s multiple comparisons test. This approach allowed for us to pinpoint specific group differences while effectively controlling the family-wise error rate.

## 3. Results

### 3.1. Soil Nutritional Status and Enzyme Activities of Cassava Rhizosphere Soil Under Different Treatments

The impact of different fertilization treatments on soil chemical properties and enzyme activities was assessed (Table 2). The OF treatment significantly increased SOM to 20.39 g/kg, which was higher than the control (CK) and RFOF treatments (*p* < 0.05). Similarly, AN levels were elevated in the OF treatment (90.57 mg/kg) compared to CK (21.99 mg/kg) and CFOF (65.50 mg/kg), indicating OF’s effectiveness in enhancing soil fertility. AP and AK availability also showed significant responses to the treatments. The OF treatment resulted in the highest available K (58.67 mg/kg) concentrations, which were significantly greater than those observed in the CK treatment (*p* < 0.05). The CFOF treatment showed intermediate effects on AP and AK, with 5.67 mg/kg and 56.33 mg/kg, respectively.

Enzyme activities varied among treatments, reflecting differences in soil biochemical properties. Neutral phosphatase activity was highest in the OF treatment (14.03 mg/d/g), followed by CFOF (11.61 mg/d/g) and RFOF (12.50 mg/d/g), with CK showing the lowest activity (3.10 mg/d/g). Sucrase and urease activities also followed a similar trend, with OF treatment showing higher activities (0.50 mg/d/g and 0.47 mg/d/g, respectively), while CK had the lowest (0.21 mg/d/g and 0.33 mg/d/g, respectively). Catalase activity was highest in the RFOF treatment (18.42 mmol/d/g), slightly higher than in OF (15.59 mmol/d/g) and CFOF (16.62 mmol/d/g), and significantly greater than in CK (9.84 mmol/d/g).

These results suggest that the application of organic fertilizer, either alone or in combination with conventional fertilizers, can significantly enhance SOM content, nutrient availability, and enzyme activities, thereby improving soil fertility and potentially plant growth.

### 3.2. Growth and Yield Parameters of Cassava

Compared to CK, the plant height, stem girth, number of tuberous roots, and yield significantly increased in OF, CFOF, and RFOF treatments (Table 3). Plant height was notably taller in the OF (234.13 cm), RFOF (237.69 cm), and CFOF (243.37 cm) treatments compared to the CK (192.67 cm) treatments, with OF, RFOF, and CFOF showing no significant difference between them. Similarly, stem girth was greater in the OF (29.75 mm), RFOF (30.31 mm), and CFOF (30.86 mm) treatments compared to the CK (24.85 mm) treatments. The number of tuberous roots was highest in the OF (10.43) and CFOF (10.60) treatments, significantly outperforming the CK (8.70) and RFOF (9.93) treatments. Yield followed a similar trend, with the OF (72.11 t ha^−1^) and CFOF (73.97 t ha^−1^) treatments yielding significantly more than the CK (52.41 t ha^−1^) and RFOF (64.68 t ha^−1^) treatments. The RFOF treatment, while not matching the yield of OF and CFOF, still outperformed the CK treatment in all measured parameters.

### 3.3. α-Diversity of Fungal Community in the Cassava Rhizosphere Soil

The assessment of alpha diversity indices revealed significant variations among the different fertilization treatments (Table 4). CK and RFOF exhibited higher values, indicating a rich and diverse fungal community. In contrast, OF and CFOF treatments showed lower diversity indices, with Chao1 values of 462.3 ± 18.53 and 385.37 ± 4.03, respectively, and ACE values of 656.03 ± 2.76 and 585.23 ± 11.54, respectively.

Simpson’s index, a measure of community evenness, showed a similar trend, with the CK treatment having the lowest value (0.033 ± 0.003), indicating a more even community, while the CFOF treatment had the highest value (0.133 ± 0.008), suggesting a less even community. The Shannon diversity index also varied, with the CK treatment having the highest value (4.63 ± 0.041) and the CFOF treatment the lowest (2.85 ± 0.05).

Good’s coverage was consistently high across all treatments, ranging from 0.995 to 0.997, indicating that the sequencing depth was sufficient to capture the majority of the fungal OTUs present in the samples.

The soil DNA sequencing outcomes are detailed in Appendix A. The rarefaction curves for each treatment group—comprising CK, OF, CFOF, and RFOF—demonstrated saturation, signifying that the sequencing depth was adequate to capture the predominant fungal diversity within each sample (Appendix A). The sequencing coverage for all treatments was high, achieving a 99% coverage rate, which assures the comprehensiveness of the detected OTUs and the reliability of the diversity indices calculated.

### 3.4. Venn Analysis of Fungal Community in the Cassava Rhizosphere Soil

The Venn diagram analysis revealed distinct OTUs across different fertilization treatments (Figure 1). The control treatment (CK) harbored a total of 427 unique OTUs, while the OF and RFOF treatments supported 119 and 316 unique OTUs, respectively. The combined treatment of CFOF showed 58 unique OTUs. Notably, 545 OTUs were shared exclusively between CK and OF treatments, and 995 were unique to CK. Similarly, 262 OTUs were unique to OF, and 561 were shared between OF and RFOF treatments. The RFOF treatment shared 517 OTUs exclusively with CFOF, and 946 OTUs were unique to RFOF. Furthermore, 979 OTUs were shared between RFOF and CK, and 304 OTUs were common to all four treatments.

These results indicate that each fertilization treatment influences the fungal community structure, promoting the presence of unique OTUs. The overlap in OTUs among treatments suggests that certain fungal taxa are responsive to multiple fertilization strategies, potentially indicating their adaptability to different soil management practices, with 183, 86, and 748 unique OTUs under OF, CFOF, and RFOF treatments, respectively (Appendix A). The number of OTUs in RFOF was lowest among these treatments.

### 3.5. Fungal Community Structure and Composition in the Cassava Rhizosphere Soil

In total, six fungal phyla were identified. Ascomycota, Basidiomycota, and Mucoromycota were the dominant phyla (>10%) under different treatments in cassava rhizosphere soil. Ascomycota was the most abundant phylum with a range from 60.21% to 89.49% in cassava rhizosphere soil (Figure 2A). The abundance of Basidiomycota, Glomeromycota, and Chytridiomycota decreased under different fertilization treatments, being especially significantly reduced in CFOF and RFOF treatments. The abundance of Mucoromycota and Mortierellomycota increased in OF treatment but was significantly reduced in CFOF and RFOF treatments.

At the genera level, *Debaryomyces*, *Aspergillus*, *Mucor*, *Arxiella*, *Strelitziana*, *Penicillium*, and *Fusarium* were the dominant in the cassava rhizosphere soil. The proportion of *Debaryomyces* and *Penicillium* significantly increased in OF, CFOF, and RFOF treatments (Figure 2B). Notably, the largest increases of *Debaryomyces* and *Penicillium* were registered under the CFOF treatments, with 154.69 and 22.51 being the abundance of folds in comparison with CK, respectively. In addition, the proportion of *Fusarium*, a genus that includes many pathogenic species causing a wide range of plant diseases and substantial economic losses, significantly decreased in OF, CFOF, and RFOF treatments, with reductions of 24.89%, 84.71%, and 76.80%, respectively. Notably, the abundance of arbuscular mycorrhizal genus *Glomus* significantly decreased among different fertilization treatments.

### 3.6. Clustering Analysis of Fungal Community in the Cassava Rhizosphere Soil

Hierarchical clustering was performed on both the samples and the fungal genera, as depicted by the dendrograms on the top and left side of the heatmap, respectively (Figure 3). Notably, the RFOF and CFOF treatments exhibit distinct fungal community profiles compared to the CK and OF treatments. The RFOF treatment is characterized by a higher relative abundance of genera such as *Mortierella*, as indicated by the red coloration in the corresponding cells. In contrast, the CK treatment shows a predominance of genera like *Debaryomyces* and *Penicillium*, represented by blue and purple hues. The OF treatment displays a fungal community composition that is more similar to the CK treatment, with a notable presence of *Fusarium* and *Talaromyces*, as shown by the intermediate coloration. The CK treatment, serving as the control, presents a baseline fungal community structure that is distinct from the fertilized treatments.

These results suggest that the application of fertilizers, particularly the organic and chemical formulations tested, significantly influences the structure and composition of fungal communities. The distinct community profiles observed in the RFOF and CFOF treatments compared to the CK and OF treatments highlight the potential for fertilizer-induced shifts in fungal community dynamics, which may have implications for soil health and ecosystem functioning.

### 3.7. Principal Coordinate Analysis in Cassava Rhizosphere Soil

To visualize the variation in fungal community structure across different treatments, we performed principal coordinate analysis (PCoA) based on the Bray–Curtis dissimilarity matrix. The PCoA results showed limited separation among samples from different treatment groups (Figure 4). PCoA1 and PCoA2 explained only 36.6% and 14.8% of the variance, respectively, accounting for a total of 51.4% of the differences among samples. This level of explained variance indicates that while the primary axes captured significant aspects of the variation, a substantial portion of the microbial community structure remained unexplained.

The distribution of points across the plot shows some degree of overlap among the groups, particularly between the CK and RFOF groups, which suggests that these treatments did not significantly alter the fungal community structure compared to the other treatments. Notably, the OF group is somewhat separated from the other groups along PCoA1, indicating a unique community structure associated with the organic fertilizer treatment. The CFOF group also shows a distinct clustering pattern, suggesting that the combination of conventional and organic fertilizers has a specific impact on the fungal community.

Overall, the PCoA results, along with the PERMANOVA (F = 1.55, *p* = 0.076), which did not detect significant differences among treatment groups, suggest that while there are observable differences in fungal community structure, these differences are not statistically significant. This could be due to high variability within groups or the influence of other unmeasured factors.

### 3.8. Correlation Analysis of Soil Chemical Properties and Enzyme Activities with Fungal Community in Cassava Rhizosphere Soil

The correlation relationship was determined by redundancy analysis (RDA). The two main coordinates accounted for 96.40% of the total variation; RDA1 and RAD2 explained 86.80% and 9.60% of the variation, respectively. Arrows in the plot represent environmental variables such as soil organic matter (SOM), available phosphorus (AP), catalase, etc. The length and direction of these arrows indicate the strength and gradient of their influence on fungal communities among different treatments. SOM, AP, AK, AN, UE, SC, CAT, and NP were negatively correlated to the RDA1 among different treatments. Additionally, UE, SC, CAT, and AP significantly influenced the structural changes in fungal communities at the genus level across different treatments, as indicated by the longest arrows (Figure 5). Overall, RDA results suggest that the application of organic fertilizer, either alone or in combination with conventional fertilizers, can significantly alter rhizosphere environmental parameters. The distinct clustering of OF and CFOF groups highlights the potential of these treatments to modify soil biochemical properties, which could have implications for soil fertility and plant growth.

The heatmap reveals correlations between soil enzyme activities, chemical characteristics, and fungal genera across fertilization treatments (Figure 6). The abundance of *Mucor* significantly positively correlated with SOM, SC, AN, and NP and negatively correlated with CAT and AK. The abundance of *Penicillium*, *Aspergillus*, and *Hortaea* significantly positively correlated with SOM, AP, AK, AN, UE, SC, CAT, and NP, displaying the highest correlation with AK. The abundance of *Fusarium* negatively correlated with, UE, CAT, and NP and positively correlated with SC. Moreover, the abundance of *Glomus* showed significant negative correlation with SOM, AP, AK, AN, UE, SC, CAT, and NP (Figure 6).

Overall, these results highlight the complex interplay between soil enzyme activities and fungal community composition under various management practices, underscoring the importance of considering microbial interactions in soil health.

## 4. Discussion

In this study, we investigated the effects of different fertilization treatments on soil quality, plant growth, and rhizosphere fungal communities in cassava cultivation. The key findings indicate that combined applications of chemical and organic fertilizers significantly enhance soil quality and promote plant growth and tuber yield compared to control treatments (Table 2). Specifically, fertilizer application increases cassava rhizosphere enzyme activity and nutrient availability, which are crucial indicators of improved soil quality (Table 1). Additionally, soil microbial diversity is altered by fertilization, with notable changes in fungal abundance and community structure (Figure 1, Figure 2, Figure 3 and Figure 4). These results highlight the importance of optimizing fertilization strategies to improve agricultural productivity and soil health.

Our findings are consistent with previous studies that demonstrate the positive impact of combined chemical and organic fertilizer applications on soil organic matter (SOM) content [10,36]. For instance, the application of chemical fertilizers in conjunction with organic fertilizers significantly increases SOM, which is vital for maintaining soil quality and promoting cassava growth and yield [36]. However, chemical fertilizers alone do not elevate SOM in cassava cultivation [11]. This suggests that organic fertilizers play a crucial role in enhancing soil organic matter content. In terms of soil microbial communities, our results show that fertilization can alter fungal abundance in the cassava rhizosphere, particularly reducing the abundance of Glomus in certain treatments (Figure 2B and Figure 3). This is likely due to the additional phosphorus supply, which reduces the dependence of plants on arbuscular mycorrhizal fungi (AMF) [37]. Previous studies have also shown that chemical fertilizers, especially nitrogen fertilizers, can diminish microbial diversity [38]. However, our study found that the treatment groups did not have a substantial impact on the overall fungal community structure, possibly due to high variability within groups or other unmeasured factors influencing microbial composition [38].

The results of this study have significant biological and practical implications for sustainable agriculture. Soil enzyme activities, such as urease, catalase, and sucrase, were found to have a substantial impact on fungal community structure (Figure 5). These enzymes are key players in nutrient cycling processes and serve as reliable indicators of soil quality [39,40]. Enhancing soil enzyme activities through fertilization can improve nutrient cycling and promote plant growth, thereby contributing to higher crop yields [36,41]. Moreover, understanding the interactions between fertilization practices and rhizosphere microbial communities can provide valuable insights for developing more sustainable farming practices. By optimizing fertilization strategies, we can maximize nutrient uptake efficiency and boost cassava yields while maintaining soil health.

Despite the significant findings of this study, there are some limitations that need to be acknowledged. Traditional sequencing methods used in this study only provide relative quantification of rhizosphere microbial communities, which limits our understanding of their absolute abundance and dynamics. Future studies should explore absolute quantification methods, such as synthetic DNA spike-in approaches, to more accurately assess the true impact of different treatments on rhizosphere microbial communities and to elucidate their ecological functions and responses. Additionally, the relatively low explained variance and non-significant PERMANOVA results suggest that other unmeasured factors may influence fungal community variation. Future research should incorporate additional metadata and use complementary statistical methods to better understand these factors. Future work should also focus on elucidating the long-term influence of diverse fertilization strategies on soil health and determining the ideal ratios and application timings of fertilizers throughout the cassava growth cycle. Examining the differential impacts of various organic and inorganic fertilizer sources, as well as their combined applications, on cassava growth and rhizosphere microbial assemblages will provide essential insights for enhancing soil health and supporting more sustainable farming practices.

## 5. Conclusions

In summary, our study demonstrates that the integration of organic and chemical fertilizers significantly enhanced cassava growth and yield compared to the control treatment (CK). The application of different fertilization strategies notably improved soil chemical properties. Despite these improvements, the overall fungal community structure remained largely unchanged, although there were significant alterations in the relative abundance of certain fungal taxa. Rhizosphere environmental parameters showed strong correlations with the fungal community structure. These findings underscore the critical role of combined organic and chemical fertilizers in promoting cassava productivity and soil health.

## Figures and Tables

**Figure 1 jof-11-00235-f001:**
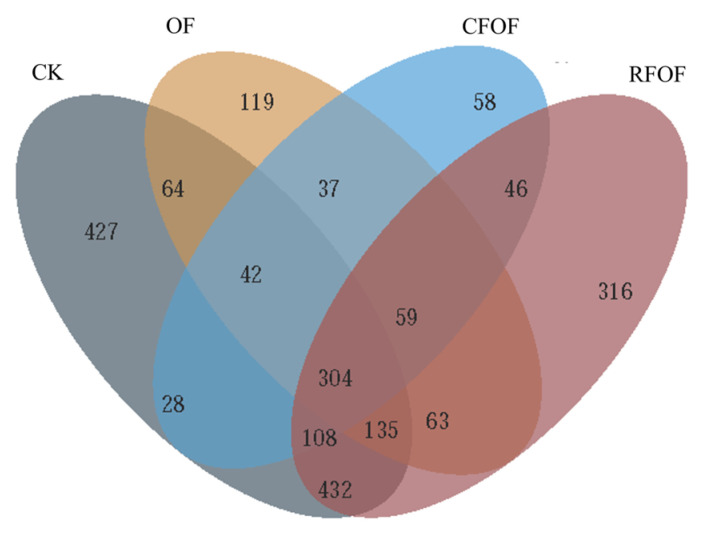
Venn diagram for shared and unique OTUs among fungal communities under different treatments.

**Figure 2 jof-11-00235-f002:**
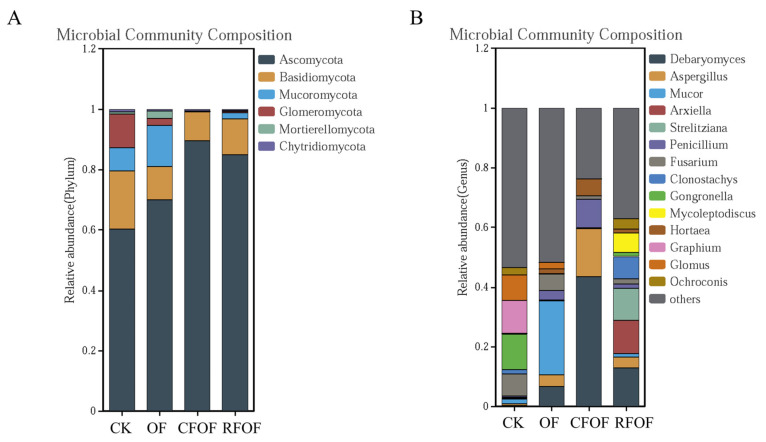
Relative abundance of the fungal phyla (**A**) and genera (**B**) in the rhizosphere of cassava under different fertilization treatments.

**Figure 3 jof-11-00235-f003:**
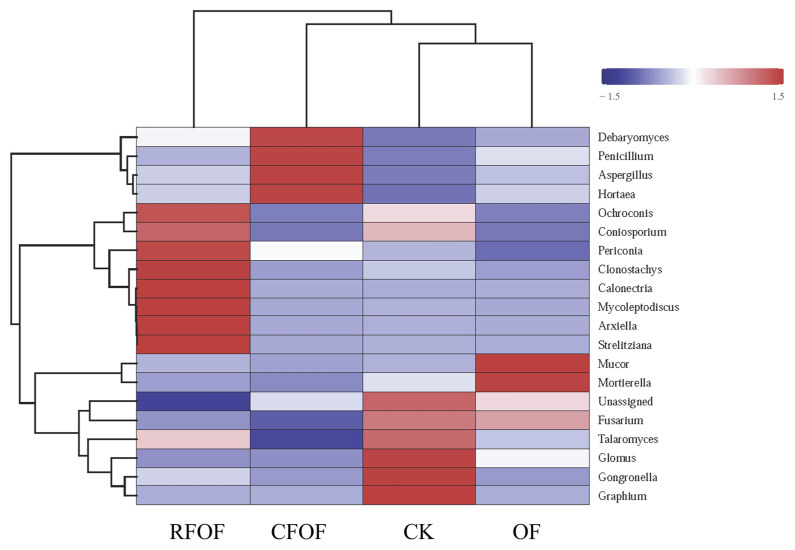
Fungal community heatmap analysis of the fungal genera detected across all the samples. The relative values for fungal genera are indicated by color intensity with the legend in the upper right of the picture. The color deepness of each block showed the fungal richness among these treatments.

**Figure 4 jof-11-00235-f004:**
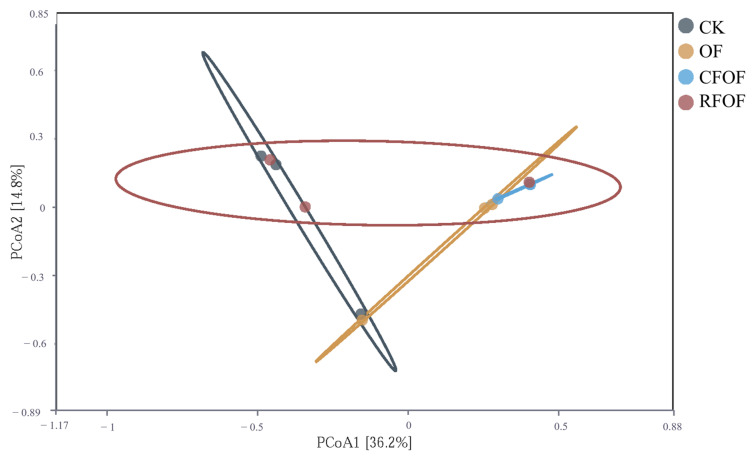
Principal coordinate analysis of OTUs across different fertilization treatments.

**Figure 5 jof-11-00235-f005:**
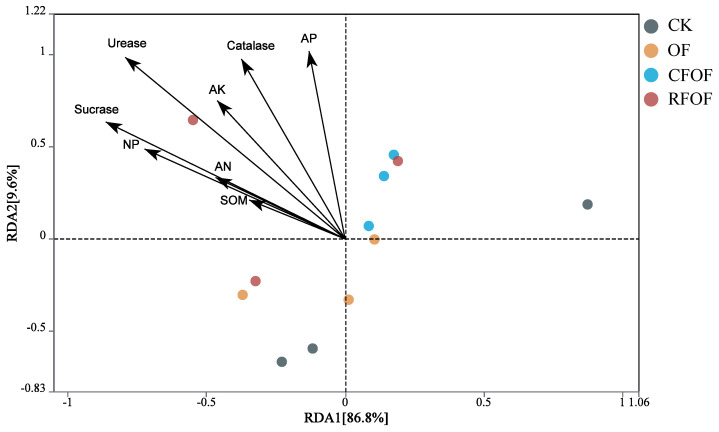
Redundancy analysis of soil chemical properties and enzyme activities with fungal community under different fertilization treatments. SOM: soil organic matter; AP, AK, and AN: available P, K, and N; NP: neutral phosphatase.

**Figure 6 jof-11-00235-f006:**
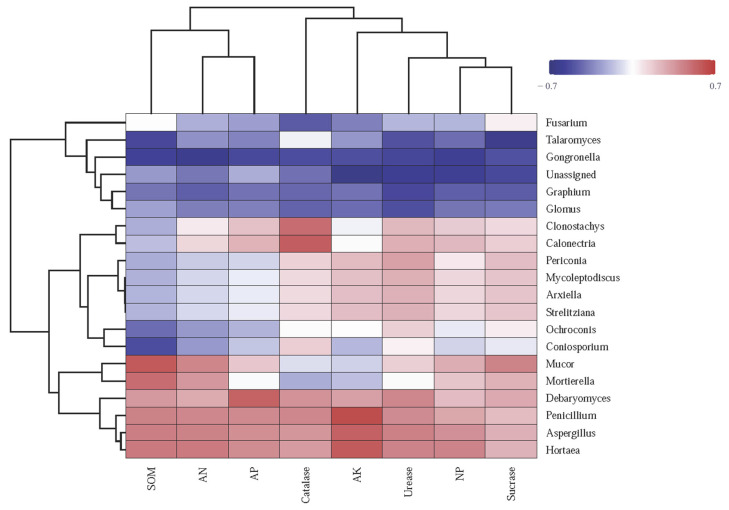
Heatmap illustrating the correlation between soil chemical properties (SOM, AP, AK, and AN), enzymatic activities (UE, SC, CAT, and NP), and the abundance of fungal genera. The relative values are indicated by color intensity with the legend in the upper right of the picture. SOM: soil organic matter; AP, AK, and AN: available P, K, and N; NP: neutral phosphatase.

**Table 1 jof-11-00235-t001:** Fertilizer application rates (kg ha^−1^) across different treatments.

Treatments	Urea (*N* > 46%)	Calcium Superphosphate (P_2_O_5_ > 16%)	Potassium Sulfate (K_2_O > 52%)	Organic Fertilizer (N+P_2_O_5_+K_2_O ≥ 5%, Organic Matter ≥ 80%, Total Humic Acid ≥ 30 g/kg, Humus ≥ 300 g/kg)
CK	0	0	0	0
OF	0	0	0	1500
CFOF	90	90	90	1500
RFOF	67.5	45	90	1500

The organic fertilizer derived from lignite fermentation. CK: no fertilizer; OF: organic fertilizer; CFOF: chemical fertilizer + organic fertilizer; RFOF: reduced chemical fertilizer + organic fertilizer.

**Table 2 jof-11-00235-t002:** The chemical characteristics and enzyme activities (means ± standard errors, *n* = 3) in cassava rhizosphere soil under different treatments.

Treatments	Soil Organic Matter (g/kg)	Available N (mg/kg)	Available P (mg/kg)	Available K (mg/kg)	Neutral Phosphatase (mg/d/g)	Sucrase (mg/d/g)	Catalase (mmol/d/g)	Urease (mg/d/g)
CK	7.80 ± 0.28 d	21.99 ± 1.50 d	3.47 ± 0.18 b	23.29 ± 1.46 b	3.10 ± 0.52 b	0.21 ± 0.05 b	9.84 ± 0.40 b	0.33 ± 0.02 b
OF	20.39 ± 0.97 a	90.57 ± 3.60 b	5.22 ± 0.50 a	58.67 ± 2.72 a	14.03 ± 1.71 a	0.50 ± 0.06 a	15.59 ± 1.59 a	0.47 ± 0.01 a
CFOF	17.59 ± 0.43 b	65.50 ± 4.61 c	5.67 ± 0.59 a	56.33 ± 3.54 a	11.61 ± 1.01 a	0.58 ± 0.06 a	16.62 ± 1.23 a	0.50 ± 0.02 a
RFOF	12.67 ± 0.57 c	61.67 ± 5.61 a	6.48 ± 0.88 a	58.33 ± 4.92 a	12.50 ± 0.68 a	0.56 ± 0.01 a	18.42 ± 1.98 a	0.50 ± 0.01 a

Different letters in a row indicate significant differences determined by Kruskal–Wallis test (*p* < 0.05).

**Table 3 jof-11-00235-t003:** Characteristics of cassava growth and yield (means ± standard errors, *n* = 10) under different treatments.

Treatments	Plant Height (cm)	Stem Girth (mm)	Number of Tuberous Roots	Yield (t ha^−1^)
CK	192.67 ± 18.27 b	24.85 ± 3.03 b	8.70 ± 0.46 c	52.41 ± 0.67 c
OF	234.13 ± 5.95 a	29.75 ± 0.33 a	10.43 ± 0.64 a	72.11 ± 0.31 a
CFOF	243.37 ± 9.14 a	30.86 ± 1.47 a	10.60 ± 0.06 a	73.97 ± 0.41 a
RFOF	237.69 ± 6.33 a	30.31 ±1.39 a	9.93 ± 0.14 b	64.68 ± 0.36 b

Note: Different letters in a row indicate significant differences determined by Kruskal–Wallis test (*p* < 0.05).

**Table 4 jof-11-00235-t004:** α-diversity indexes of fungal community in the cassava rhizosphere (means ± standard errors, *n* = 3) under different treatments.

Treatments	Chao1	ACE	Simpson	Shannon_e	Goods_coverage
CK	938.8 ± 1.54 a	1061.06 ± 8.93 a	0.033 ± 0.003 c	4.63 ± 0.041 a	0.995
OF	462.3 ±18.53 b	656.03 ± 2.76 b	0.069 ± 0.011 b	4.19 ± 0.032 b	0.996
CFOF	385.37 ± 4.03 c	585.23 ± 11.54 c	0.133 ± 0.008 a	2.85 ± 0.05 c	0.997
RFOF	963.5 ± 29.70 a	1102.47 ± 57.74 a	0.118 ± 0.024 a	4.15 ± 0.21 b	0.997

Note: Different letters in a row indicate significant differences determined by Kruskal–Wallis test (*p* < 0.05).

## Data Availability

The complete datasets generated in our study have been submitted in the NCBI Sequence Read Archive database (BioProject and accession: PRJNA1152878).

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
