# Peer review of "The Effect of Different Types of Fertilizers on the Growth of Cassava and the Fungal Community in Rhizosphere Soil"

_jof, 2025, doi:10.3390/jof11030235_

Round 1

Reviewer 1 Report (Previous Reviewer 1)

Тhe corrected version of the manuscript is clearer and provide important insights into how different types of fertilizers influence on the growth of cassava and the fungal community in the rhizosphere soil.

It is very interesting and useful that the authors have investigated the effect of different types of fertilizers influence on the growth of cassava and the fungal community in the rhizosphere soil . The authors adopted the recommendations and thus significantly improved the way in which information presented.

In general, revised form of manuscript was written well. The objectives of the study were clearly presented 

The material and methods used in this study as well as the research results are adequately described.

In section Discussion, the authors analyze their results and connect them with the literature data. The conclusions were based on the data of this study.

Author Response

Comment: The corrected version of the manuscript is clearer and provide important insights into how different types of fertilizers influence on the growth of cassava and the fungal community in the rhizosphere soil.

It is very interesting and useful that the authors have investigated the effect of different types of fertilizers influence on the growth of cassava and the fungal community in the rhizosphere soil.

The authors adopted the recommendations and thus significantly improved the way in which information presented. In general, revised form of manuscript was written well. The objectives of the study were clearly presented.

The material and methods used in this study as well as the research results are adequately described. In section Discussion, the authors analyze their results and connect them with the literature data. The conclusions were based on the data of this study.

A: We are very grateful for your positive feedback on the revised version of our manuscript. Your comments are highly encouraging and validate the efforts we have made to address the previous concerns and improve the overall quality of our work.

Once again, thank you for your constructive and supportive review.

Reviewer 2 Report (New Reviewer)

The study presents interesting findings and contributes valuable insights to the field. However, the manuscript lacks sufficient details and interpretation in key sections, which makes it difficult for readers to fully understand the significance of the results.

For instance, in several sections, the results are presented without adequate discussion or interpretation, leaving the reader to draw conclusions. Additionally, certain methodological details need further clarification to ensure the study’s reproducibility. 

Abstract

Introduction:

The introduction is very general and lacks specific literature on particular products and hosts.

The aim of the study must be stated clearly. 

Materials and methods:

Lacks of proper structure:  

1. Create a table to clearly present the specifications for CF (Chemical Fertilizer) and OF (Organic Fertilizer).

2. Specify the exact amount of each fertilizer used per plant.

3. Define organic fertilizer by listing its ingredients.

4. Explain what SC12 is?

5. Clearly state the number of plants used per treatment and the number of replications.

6. There were 12 plots, but only 10 were sampled—clarify the reason for this selection and confirm if samples were randomly taken from each plot.

7. While referring to another study for analyses is acceptable, details of the analyses performed must be explicitly stated.

8. Describe how the growth parameters were measured.

9. The statistical analysis section is too brief—provide more details on the methods used. 

Results: 

The results lack of details and they have been referred to tables without explanation. 

In section 3.6. the authors have merely described the colors used for analysis instead of providing an interpretation of their significance.

Discussion: 

The authors have confused the discussion with the results. In the discussion section, data should be interpreted and contextualized rather than simply described. References to tables and figures should be limited to the results section, not the discussion.

The current structure of the discussion is not acceptable. I recommend following the structure below:

Paragraph 1. Summary of Key Findings – Briefly highlight the most important results.

Paragraph 2. Comparison with Existing Literature – Relate findings to previous studies, highlighting similarities, differences, and possible explanations.

Paragraph 3. Biological/Practical Implications – Discuss the significance of the findings in a broader context.

Paragraph 4. Limitations and Future Directions – Acknowledge any limitations and suggest areas for further research.

In general this is the structure for each paragraph: 

·       Start the paragraph with a topic sentence

·       Outcome of this study

·       Comparing with the other studies

·       One sentence of conclusion

Author Response

Q: Major comments

The study presents interesting findings and contributes valuable insights to the field. However, the manuscript lacks sufficient details and interpretation in key sections, which makes it difficult for readers to fully understand the significance of the results.

For instance, in several sections, the results are presented without adequate discussion or interpretation, leaving the reader to draw conclusions. Additionally, certain methodological details need further clarification to ensure the study’s reproducibility.

R: We sincerely thank you for your positive evaluation of our study and your constructive feedback. Your comments have been extremely helpful in identifying areas where our manuscript could be improved, particularly in terms of detailed discussions and methodological clarity. We have carefully reviewed your suggestions and have made substantial revisions to address these concerns comprehensively.

Detail comments

Abstract

Q: Introduction:

The introduction is very general and lacks specific literature on particular products and hosts.

The aim of the study must be stated clearly. 

A: Thank you for your insightful comment. In the introduction, we have provided a broad overview of soil health, the challenges associated with fertilizer use, and the importance of cassava as a staple crop. These contents are essential for setting the stage for our research and highlighting the significance of sustainable fertilization practices. However, we understand your point that the introduction could benefit from more specific literature on particular products (e.g., specific types of organic fertilizers) and hosts (i.e., cassava as the host plant). We revised the introduction to include more detailed discussions on these aspects. This will help to further contextualize our study and strengthen the relevance of our research objectives. Thank you again for your valuable feedback.

Materials and methods:

Lacks of proper structure:  

Q: 1. Create a table to clearly present the specifications for CF (Chemical Fertilizer) and OF (Organic Fertilizer).

A: Thank you for your suggestion. We have created a new table (Table 1) to clearly present the specifications for both Chemical Fertilizer (CF) and Organic Fertilizer (OF) used in our study. This table includes detailed information on the nutrient content and other relevant specifications for each type of fertilizer, as well as the specific application rates used in our experimental treatments. We believe this addition will enhance the clarity and transparency of our experimental design. Thank you again for your valuable feedback.

Q: 2. Specify the exact amount of each fertilizer used per plant.

Thank you for your suggestion. We have now specified the exact amount of each fertilizer applied per plant in our study. The details are provided in Table 1 and further explained in Section 2.2 (line 122-126). We believe this addition will provide a clearer understanding of the fertilizer application in our study.

Thank you again for your valuable feedback.

Q: 3. Define organic fertilizer by listing its ingredients.

A: Thank you for your insightful comments. In response to your request to define the organic fertilizer by listing its ingredients, we have already provided a detailed description of the organic fertilizer in Section 2.1 "Description of study site and materials," specifically on lines 104-106. Here, we clearly listed the key components of the organic fertilizer, including its nutrient content and organic matter composition. We believe this information provides a comprehensive definition of the organic fertilizer used in our study. Thank you again for your valuable feedback.

Q: 4. Explain what SC12 is?

A: Thank you for raising this question. SC12 is a cassava cultivar, and its full name and description are provided in lines 101-102 of the manuscript.

Q: 5. Clearly state the number of plants used per treatment and the number of replications.

A: Thank you for your suggestion. We appreciate your attention to the clarity of our experimental design. We would like to confirm that the number of plants per treatment and the number of replications has already been detailed in our manuscript (line 116-119). Here, each experimental plot contains 36 plants (arranged in 6 rows with 6 plants per row, spaced at 1.0 m × 1.0 m). Each treatment was replicated three times. There are 12 experimental plots in total (3 replications × 4 treatments). We believe this information provides a clear and comprehensive overview of our experimental design. If further clarification is needed, please let us know. Thank you again for your valuable feedback.

Q: 6. There were 12 plots, but only 10 were sampled—clarify the reason for this selection and confirm if samples were randomly taken from each plot.

A: Thank you for your query regarding the sampling procedure. We appreciate the opportunity to clarify this aspect of our study.

We would like to confirm that all 12 experimental plots were sampled and measured. This information has already been described in our manuscript (line 130). Specifically, within each plot, we randomly selected 10 plants out of the 36 plants for growth data measurement and collected rhizosphere soil from these plants for subsequent analyses. These 10 sampling points were chosen randomly to ensure a representative and unbiased assessment of plant growth and soil properties within each plot. We believe this approach provided a comprehensive and robust dataset for our analyses.

Thank you again for your valuable feedback. We are happy to provide additional clarification if needed.

Q: 7. While referring to another study for analyses is acceptable, details of the analyses performed must be explicitly stated.

A: Thank you for your valuable feedback. We appreciate your suggestion to provide more explicit details of the analyses performed. In response, we have now included a detailed description of the methods used to measure soil enzyme activities in Section 2.4 (" Soil enzyme activities detection ") of our manuscript (line 151-163). We believe this additional information will enhance the clarity and reproducibility of our methods. Thank you again for your constructive feedback.

Q: 8. Describe how the growth parameters were measured.

A: Thank you for your comment. We have now provided a detailed description of how the growth parameters were measured in our study. This information has been added to Section 2.3 ("Measurement of Growth and Yield Parameters") of our manuscript. We believe this detailed description will provide a clear understanding of our methodology. Thank you again for your valuable feedback.

Q: 9. The statistical analysis section is too brief—provide more details on the methods used. 

A: Thank you for your valuable feedback regarding the statistical analysis section. We appreciate your suggestion to provide more details on the methods used. In response, we have expanded and clarified the statistical analysis section in our manuscript to include more comprehensive information on the methods employed (line 215-223). We believe that these revisions provide a more detailed and transparent account of the statistical methods used in our study, enhancing the clarity and reproducibility of our research. Thank you again for your constructive feedback.

Results: 

Q: The results lack of details and they have been referred to tables without explanation. 

In section 3.6. the authors have merely described the colors used for analysis instead of providing an interpretation of their significance.

A: Thank you for your valuable feedback regarding the clarity and interpretation of our results. We have carefully reviewed your comments and have made the necessary revisions to address the concerns raised. We appreciate your constructive feedback and believe that these revisions have significantly enhanced the clarity and interpretability of our results. Thank you again for your guidance.

Discussion: 

Q: The authors have confused the discussion with the results. In the discussion section, data should be interpreted and contextualized rather than simply described. References to tables and figures should be limited to the results section, not the discussion.

The current structure of the discussion is not acceptable. I recommend following the structure below:

Paragraph 1. Summary of Key Findings – Briefly highlight the most important results.

Paragraph 2. Comparison with Existing Literature – Relate findings to previous studies, highlighting similarities, differences, and possible explanations.

Paragraph 3. Biological/Practical Implications – Discuss the significance of the findings in a broader context.

Paragraph 4. Limitations and Future Directions – Acknowledge any limitations and suggest areas for further research.

In general this is the structure for each paragraph: 

  •      Start the paragraph with a topic sentence
  • Outcome of this study
  • Comparing with the other studies
  • One sentence of conclusion

A: Thank you for your feedback. We have carefully reviewed your comments and have revised the discussion section accordingly to ensure it aligns with your recommendations. We have restructured the discussion to clearly separate the presentation of results from their interpretation and contextualization. The revised section now follows the suggested format, with distinct paragraphs summarizing key findings, comparing our results with existing literature, discussing the broader implications, and acknowledging limitations while suggesting future directions. We appreciate your guidance and believe these changes have significantly improved the clarity and coherence of our manuscript.

Reviewer 3 Report (New Reviewer)

Dear authors

I have reviewed the Manuscript ID: jof-3468223
title: “The Effect of Reduced Chemical Fertilizer Combined with Organic Fertilizer on the Growth of Cassava and the Fungal Community in the Rhizosphere Soil”.

I consider that the work is relevant because it addresses a very current subject. The article is in line with the current needs of agriculture to increase yield due to the increasing demand for food. Also, are interesting study strategies with the aim to reduce chemical fertilizer application to maintain the soil health. However, I consider that the work has some weaknesses in the presentation of the objectives which could be easily clarified.

According to mentioned above I consider that the manuscript could be accept to publish in Journal of Fungi, after modifications.

Dear  authors

I have reviewed the Manuscript ID: jof-3468223
title: “The Effect of Reduced Chemical Fertilizer Combined with Organic Fertilizer on the Growth of Cassava and the Fungal Community in the Rhizosphere Soil”.

I consider that the work is relevant because it addresses a very current subject. The article is in line with the current needs of agriculture to increase yield due to the increasing demand for food. Also, are interesting study strategies with the aim to reduce chemical fertilizer application to maintain the soil health. However, I consider that the work has weaknesses in the presentation of the objectives.

According to mentioned above I consider that the manuscript could be accept to publish in Journal of Fungi, after modifications.

I find differences between the way the research objective is presented in the Introduction, and in the Abstract. I suggest clarifying the priority order of the objective (s) in the manuscript to present them in the Introduction.

In the Abstract: According to the authors: The objectives were: to investigate the effects of three fertilizer treatments-organic fertilizer (OF), chemical fertilizer combined with organic fertilizer (CFOF), and reduced chemical fertilizer combined with organic fertilizer (RFOF)-on the fungal community structure, chemical properties (SOM, AP, AN and AK) and enzyme activities (NP, SC, CAT and UE) in cassava rhizosphere.

In the Introduction:  The objective of this study was to investigate the impact of reduced chemical fertilizer application in conjunction with organic fertilizers on fungal abundance and community composition in cassava rhizosphere. The interrelationships between soil chemical properties, soil fungal communities and plant growth parameters were elucidated by means of 85 field experiments and molecular analyses”.

On the other hand, according to the objectives proposed in the Introduction the main objective seems to be focused on the study of the impact of fertilizers on the fungal community composition while in Conclusions it does not seem to be the topic most important about the work.

Additionally, in the highlighted sentence they do not mention the study of enzymatic activity.

I am listed below other suggestions and comments that I think should be made in the text to improve or clarify the manuscript for publication.

Other corrections have been made in the pdf text.

 -Lines 54-55: fertilizers are still required because nutrients may become insufficient in soil.

Dot is missing, and it should start with a capital letter.. Fertilizers….

-Lines 113-114:  Cassava rhizosphere soil samples were collected in March 202, following the methods described by Cai et al.

Check the year and add the dot at the end of the sentence.

-Lines 99-123: 2.2. Experimental design and sample collection

It is not mentioned in this section that a portion was used to analyze the fungal community.

-Lines 132-133: The determination of soil enzyme activities was performed using the colorimetric methods proposed by Gao et al. [33].

Add dot at the end of the sentence.

-Lines 139-140: …”Stem girth, plant height, number of tuberous root and yield were detected and analyzed as described by Cai et al. [11].”…..

I think the word “registered” is more appropriate, not “detected”.

-Lines 190-191: Table 2: characteristics of cassava growth and yield (means ± standard errors, n=3) under different 190 treatments.

“C”

-Lines 231-233: Notably, the largest increases of Debaryomyces and Penicillium were registered under the CFOF treatments, with 154.69 and 22.51 the abundance of folds in comparison with CK, respectively.

Add comma.

I consider the word 'registered' more appropriate that “detected”.

Lines 250-251: In this sentence: …” Above all, the richness of mainly beneficial fungi was increased, and the richness of some harmful fungi was decreased”…..

This sentence is not clear. First, because it is not clarified what is being referred to when it mentions that some beneficial fungi are increased, in comparison to the control or among the treatments regarding RFOF? And, what is the authors' criteria for considering species as beneficial or harmful? I consider it should be clarified in the manuscript which species are being referred to."

Lines 283-284: Figure 5. redundancy analysis of soil chemical properties, enzyme activities with fungal community under different fertilization treatments.

“R”. I suggest adding the references of the Figure.  For example: AN, AK, AP…

Lines 363-365: Environmental parameters, particularly UE, SC, CAT, and AP, showed strong correlations with fungal community structure.

I would add rhizosphere environmental parameters….

Author Response

Q: I find differences between the way the research objective is presented in the Introduction, and in the Abstract. I suggest clarifying the priority order of the objective (s) in the manuscript to present them in the Introduction.

In the Abstract: According to the authors: The objectives were: to investigate the effects of three fertilizer treatments-organic fertilizer (OF), chemical fertilizer combined with organic fertilizer (CFOF), and reduced chemical fertilizer combined with organic fertilizer (RFOF)-on the fungal community structure, chemical properties (SOM, AP, AN and AK) and enzyme activities (NP, SC, CAT and UE) in cassava rhizosphere.

In the Introduction:  The objective of this study was to investigate the impact of reduced chemical fertilizer application in conjunction with organic fertilizers on fungal abundance and community composition in cassava rhizosphere. The interrelationships between soil chemical properties, soil fungal communities and plant growth parameters were elucidated by means of 85 field experiments and molecular analyses”.

On the other hand, according to the objectives proposed in the Introduction the main objective seems to be focused on the study of the impact of fertilizers on the fungal community composition while in Conclusions it does not seem to be the topic most important about the work.

A: Response: Thank you so much for your careful review and valuable comments. We fully recognize the importance of consistency and clarity in presenting our research objectives, and we appreciate your pointing out the discrepancies between the Introduction and the Abstract, as well as the need to better align the conclusions with the stated objectives. Thank again for your input.

Q: -Lines 54-55: fertilizers are still required because nutrients may become insufficient in soil.

Dot is missing, and it should start with a capital letter.. Fertilizers….

A: Thank you for your attention to detail regarding the punctuation and capitalization in lines 54-55. We have revised the sentence to ensure that it begins with a capital letter, as you suggested (lines 54-55).

Regarding the placement of the period, we have carefully reviewed the sentence and confirmed that it adheres to the journal's requirements. The period is correctly placed at the end of the sentence (behind the reference number), following the journal's guidelines for punctuation. We have also conducted a thorough review of the entire manuscript to ensure consistency and accuracy in punctuation and capitalization throughout. Thank you again for your valuable feedback. We appreciate your guidance in helping us improve the quality of our manuscript

Q:-Lines 113-114:  …Cassava rhizosphere soil samples were collected in March 202, following the methods described by Cai et al.

Check the year and add the dot at the end of the sentence.

A: Thank you for your careful attention to detail. We apologize for the typographical error in the year mentioned in lines 113-114. We have corrected the year from "202" to "2023" to accurately reflect the date when the cassava rhizosphere soil samples were collected (line 128). Regarding the placement of the period, we have confirmed that it is correctly positioned at the end of the sentence, following the citation number [11], which followed the citation style used in the articles previously published in this journal. Thank you again for your valuable feedback. We appreciate your guidance in helping us improve the quality of our manuscript.

Q: -Lines 99-123: 2.2. Experimental design and sample collection

It is not mentioned in this section that a portion was used to analyze the fungal community.

A: Thank you for your careful reading and valuable comments. In response to your concern, we would like to clarify that the fungal community analysis was indeed mentioned in the section. Specifically, we stated that “Another portion was placed in dry ice for subsequent high-throughput sequencing analysis.” (lines136-137). The fungal community analysis was based on the data obtained from this high-throughput sequencing. To further improve the clarity, we will add a sentence to explicitly state that the fungal community analysis was conducted using the data from the high-throughput sequencing of the portion placed in dry ice. This will help to avoid any confusion for future readers. Thank you again for your insightful feedback.

Q: -Lines 132-133: The determination of soil enzyme activities was performed using the colorimetric methods proposed by Gao et al. [33].

Add dot at the end of the sentence.

Response: Thank you for your attention to the citation format in our manuscript. As we mentioned above, we have followed the citation style used in the articles previously published in this journal. Specifically, the journal’s style guide requires that only a single period is used after the citation number, without an additional period after “et al.” Therefore, we have adopted this format consistently throughout our manuscript to align with the journal’s requirements.

Q: -Lines 139-140: …”Stem girth, plant height, number of tuberous root and yield were detected and analyzed as described by Cai et al. [11].”…..

I think the word “registered” is more appropriate, not “detected”.

Response: Thank you for your insightful comment regarding the terminology used in our manuscript. We appreciate your suggestion to replace the word "detected" with "registered" in the sentence. Upon reflection, we agree that "registered" is a more precise term in this context. "Detected" might imply a more passive observation, whereas "registered" clearly indicates that these parameters were actively measured and recorded. Moreover, to keep the manuscript consistent, the dot before reference number was removed (line 166).

We appreciate your careful reading and valuable feedback, which helps us improve the accuracy and clarity of our manuscript. Thank you again for your time and suggestions.

Q: -Lines 190-191: Table 2: characteristics of cassava growth and yield (means ± standard errors, n=3) under different 190 treatments.

“C”

A: Thank you very much for pointing out the writing errors in our manuscript. We have made the necessary revisions to enhance the overall quality of the paper. (line 267)

Q: -Lines 231-233: Notably, the largest increases of Debaryomyces and Penicillium were registered under the CFOF treatments, with 154.69 and 22.51 the abundance of folds in comparison with CK, respectively.

Add comma.

I consider the word 'registered' more appropriate that “detected”.

A: Thank you for your valuable comments. We have revised the manuscript accordingly. Specifically, we have added a comma in the sentence on Line 331.

Q: Lines 250-251: In this sentence: …” Above all, the richness of mainly beneficial fungi was increased, and the richness of some harmful fungi was decreased”…..

This sentence is not clear. First, because it is not clarified what is being referred to when it mentions that some beneficial fungi are increased, in comparison to the control or among the treatments regarding RFOF? And, what is the authors' criteria for considering species as beneficial or harmful? I consider it should be clarified in the manuscript which species are being referred to."

Response: Thank you for your critical feedback on the clarity of our conclusions regarding beneficial and harmful fungi. Given that our study focused only on the genus level, the terms “beneficial” and “harmful” can indeed be ambiguous at this taxonomic level. To address this ambiguity, we have revised sentences in question. We believe this revision clarifies our findings and avoids potential misinterpretation. We hope these changes address your concerns and enhance the clarity and accuracy of our manuscript. Thank you again for your valuable input.

Q: Lines 283-284: Figure 5. redundancy analysis of soil chemical properties, enzyme activities with fungal community under different fertilization treatments.

“R”. I suggest adding the references of the Figure.  For example: AN, AK, AP…

Response: Response: Thank you for your valuable suggestions. We have addressed the comments by adding relevant references, and correcting capitalization issues (line 404). These changes have enhanced the clarity and quality of the manuscript. We appreciate your feedback and your efforts in reviewing our work.

Q: Lines 363-365: Environmental parameters, particularly UE, SC, CAT, and AP, showed strong correlations with fungal community structure.

would add rhizosphere environmental parameters….

A: Thank you for your suggestion regarding the terminology used to describe environmental parameters. We agree that “rhizosphere environmental parameters” is more precise and accurate. We have revised the manuscript to reflect this change (line 482). This adjustment enhances the clarity and scientific accuracy of our discussion. Thank you again for your valuable input.

This manuscript is a resubmission of an earlier submission. The following is a list of the peer review reports and author responses from that submission.

Round 1

Reviewer 1 Report

Dear editor,

First of all, you should add line numbers in the manuscript, because it is easier to review and suggest changes

Abstract contains basic informations about this topic, enough to introduce the reader to the topic of the manuscript.

It would be better if the authors rearranged the structure of the Introduction section. This refers to the first two paragraphs in this part. For example the first sentence in this part is not well written, so it needs to be rearranged.

The part Material and methods can be more clearly stated, concretely>

Conventional fertilizer is chemical fertilizer?

Instead N 90 kg/ha should be written N 90kg ha-1.  In this way and further in the manuscript.

I would ask the authors to clarify whether N 90 kg/ha, P2O5 90 kg/ha and K2O 90 kg/ha are three individual fertilizer or it is a compound and represents the complex structure of one fertilizer?

What is organic fertilizer by structure? Manure, green manure?

I found interesting informations in the section Results is well structured and contains all the results. They are clearly presented in graphs and tables.

In Discussion part authors are describe and analyzes their results and linked them with already published literature.

The conclusions in the section showed that different fertilization strategies improve the growth of cassava and yield compared to control variant. The authors indicate to the importance of management practices for increasing productivity in agriculture and that therefore research on this topic should be continued in the future.

All my comments are contained in the previous - Major comments and Deatlc omments are written as a whole.

Reviewer 2 Report

The paper is devoted to the ecologically and economically interesting and important topic. However, the unacceptable quality of the results' presentation and interpretation does not allow considering the paper for publication in a scientific journal.

Abstract and hereafter in the text: the authors did not measure any soil physical parameters (color, texture, structure, porosity, density, consistence, aggregate stability, and temperature), only chemical parameters. Thus, the word "physico" must be deleted.

Introduction, the second paragraph – it is unclear whether the authors speak about the mycorrhizal associations. The last paragraph – the study objectives should be stated more clearly (see the attached PDF file).

Material and Methods

Sequencing data processing and statististical analysis are more than insufficiently described. Nothing is written about the indexes of alpha diversity, how beta diversity was determined, how Venn diagram was constructed, how clustering, PCoA, and correlation analysis were performed.

Results

Concerning alpha diversity indexes – the expression "index of cassava rhizosperic soil" is just a nonsense, because these indexes characterize the communities and not the soil.

 The definition and description of cluster analysis are not right at all. Cluster analysis is used to determine the similarity of fungal communities' composition (beta diversity) and is based on the abundance of fungal units.

PCoA (as well as the correlation analysis) is just meaningless at the phylum level, because Ascomycota and Basidiomycota are the most diverse and abundant phyla in the fungal kingdom and dominate fungal communities in almost all types of soil.

In general, the section contains the information, which should be placed either in M@M or in Discussion (see the attached PDF file).

Discussion – in fact, it is absent, because a little bit more than one page of this section consists mostly of the results' repetitions, the general or unclear statements, and some citations of the previous studies. Moreover, the last paragraph could be written without performing the presented study.

Conclusion - it is absolutely unclear from the text how the authors distinguished between beneficial and harmful fungi.

The language of the manuscript  is beyond any grammatical, stylistical, and scientific norms (see the attached PDF file).

Because the text does not contain line numbering, the majority of comments, corrections, and suggestions are inserted into the attached PDF version of manuscript. 

Reviewer 3 Report

The authors of the study touched upon the relevant and important topic of the influence of various combinations of organic and mineral fertilizers on the physicochemical properties of soils and the composition of the fungal community. However, they failed to consider it impartially, and they largely passed off the desired results as real.

Due to the application of fertilizers under already vegetating cassava plants (clearly uneven), the correctness of the selection of soil samples and the representativeness of the samples remain questionable.

The groups of fungi that the authors classify as useful (ascomycetes, basidiomycetes) contain many pathogens of agricultural plants. Thus, the well-known pathogen Rhizoctonia solani belongs to basidiomycetes.

Lines 21-22: “The application of diverse fertilizer treatments resulted in enhanced plant growth and yields, due to notable enhancements in soil physicochemical properties”. To change the physicochemical properties of soils, either gigantic quantities of fertilizers or their long-term use are required. We only see a change in chemical properties (NPK content) and an increase in SOM content, which is obviously the result of incorrect sampling.

Line 24: the conclusion about an increase in the content of beneficial fungi and a decrease in harmful ones is not supported by the results.

Lines 42-43: Please provide references to works where pH, organic matter content and nutrient content are included among soil health indicators.

Lines 72-73: The active substances in both mineral and organic fertilizers are the same, these are chemical compounds, mainly nitrogen, phosphorus and potassium. Therefore, there can be no talk of "organic fertilizers release", this is either a misunderstanding of the processes taking place, or an incorrect phrase.

Line 173: Why was the Kruskal-Wallis test chosen?

Table 1: I recommend once again explaining the abbreviated names of fertilizer options in the legend

How was the organic matter content determined in the absence of soil mixing after applying fertilizers? In the places of application?

What are the properties of the applied organic fertilizers? Their source? How much NPK do they have?

Line 200: remove the dot after "alpha"

Lines 243-245: Explain the mechanism for reducing pathogenic fungi and increasing beneficial ones. How do fertilizers "recognize" beneficial and harmful fungi?

There are a large number of pathogens among both Aspergilli and Penicillium.

How much organic matter should be added per hectare to double its amount?

Thus, the authors' conclusions are absolutely not supported by the results. The manuscript needs to be rewritten, making objective conclusions, and not those that the authors need.

The authors of the study touched upon the relevant and important topic of the influence of various combinations of organic and mineral fertilizers on the physicochemical properties of soils and the composition of the fungal community. However, they failed to consider it impartially, and they largely passed off the desired results as real.

Due to the application of fertilizers under already vegetating cassava plants (clearly uneven), the correctness of the selection of soil samples and the representativeness of the samples remain questionable.

The groups of fungi that the authors classify as useful (ascomycetes, basidiomycetes) contain many pathogens of agricultural plants. Thus, the well-known pathogen Rhizoctonia solani belongs to basidiomycetes.

Lines 21-22: “The application of diverse fertilizer treatments resulted in enhanced plant growth and yields, due to notable enhancements in soil physicochemical properties”. To change the physicochemical properties of soils, either gigantic quantities of fertilizers or their long-term use are required. We only see a change in chemical properties (NPK content) and an increase in SOM content, which is obviously the result of incorrect sampling.

Line 24: the conclusion about an increase in the content of beneficial fungi and a decrease in harmful ones is not supported by the results.

Lines 42-43: Please provide references to works where pH, organic matter content and nutrient content are included among soil health indicators.

Lines 72-73: The active substances in both mineral and organic fertilizers are the same, these are chemical compounds, mainly nitrogen, phosphorus and potassium. Therefore, there can be no talk of "organic fertilizers release", this is either a misunderstanding of the processes taking place, or an incorrect phrase.

Line 173: Why was the Kruskal-Wallis test chosen?

Table 1: I recommend once again explaining the abbreviated names of fertilizer options in the legend

How was the organic matter content determined in the absence of soil mixing after applying fertilizers? In the places of application?

What are the properties of the applied organic fertilizers? Their source? How much NPK do they have?

Line 200: remove the dot after "alpha"

Lines 243-245: Explain the mechanism for reducing pathogenic fungi and increasing beneficial ones. How do fertilizers "recognize" beneficial and harmful fungi?

There are a large number of pathogens among both Aspergilli and Penicillium.

How much organic matter should be added per hectare to double its amount?

Thus, the authors' conclusions are absolutely not supported by the results. The manuscript needs to be rewritten, making objective conclusions, and not those that the authors need.